# Evaluation of the *Astragalus exscapus* L. subsp. *transsilvanicus* Roots’ Chemical Profile, Phenolic Composition and Biological Activities

**DOI:** 10.3390/ijms232315161

**Published:** 2022-12-02

**Authors:** Katalin Szabo, Floricuta Ranga, Simon Elemer, Rodica Anita Varvara, Zorita Diaconeasa, Francisc Vasile Dulf, Dan Cristian Vodnar

**Affiliations:** 1Faculty of Food Science and Technology, University of Agricultural Sciences and Veterinary Medicine, 400372 Cluj-Napoca, Romania; 2Technological Transfer Center COMPAC, University of Agricultural Sciences and Veterinary Medicine, 400372 Cluj-Napoca, Romania; 3Faculty of Agriculture, Department of Environmental and Plant Protection, University of Agricultural Sciences and Veterinary Medicine, Calea Manastur 3-5, 400372 Cluj-Napoca, Romania

**Keywords:** *Astragalus memebranaceus*, Astragaloside IV, B16F10 melanoma cells, LC-MS, minimum inhibitory concentration (MIC)

## Abstract

Novel and natural molecules for pharmaceutical applications are a contemporary preoccupation in science which prompts research in underexplored environments. *Astragalus exscapus* ssp. *transsilvanicus* (Schur) Nyár. (*ASTRA*) is a plant species endemic to Transylvania, having a very similar root system with that of *A. membranaceus* (Fisch.) Bunge, known for its health promoting properties. The present study endeavored to perform basic characterization of the *ASTRA* roots by proximate analysis, to investigate the fatty acid profile of the lipids extracted from the *ASTRA* roots, to examine the phenolic composition of the root extracts by liquid chromatography, and to evaluate the biological activities through determination of the antioxidant, antimicrobial and cytotoxic capacities of the extracts. The primary compounds found in the *ASTRA* roots were carbohydrates and lipids, and the fatty acid composition determined by GC-MS showed linoleic acid as preponderant compound with 31.10%, followed by palmitic, oleic and α-linolenic acids with 17.30%, 15.61% and 14.21%, respectively. The methanol extract of the *ASTRA* roots presented highest phenolic content, Astragaloside IV being the predominant compound with 425.32 ± 0.06 µg/g DW. The antimicrobial assay showed remarkable antimicrobial potential of the extract at a concentration of 0.356 and 0.703 mg *ASTRA* root powder (DW)/mL, highlighting its efficacy to inhibit *S. aureus* and *S. epidermidis* bacterial strains. Furthermore, the cell proliferation assessment showed the noteworthy proficiency of the treatment in inhibiting the proliferation of B16F10 melanoma cells.

## 1. Introduction

Plant-derived bioactive molecules have gained substantial interest in the prevention and/or treatment of different non-communicable diseases. Numerous studies have been conducted on secondary metabolites such as flavonoids, carotenoids, and other natural compounds of plant origin indicating their health benefits in epidemiologic, clinical, and experimental trials [1]. In the context of antibiotic resistance, which is a major problem in modern medicine, and the increasing world population, the urgent need to find novel and natural molecules for pharmaceutical applications is prompting research in underexplored environments.

Stemless milkvetch (*Astragalus exscapus* L.) is a little-known plant species from the *Astragalus* genus, *Fabaceae* family, studied mostly from an ecological point of view for conservational purposes [2]. *A. exscapus* is a long-lived perennial herb native to dry grasslands of Central Europe, with small populations and a relatively high habitat specificity [3]. This remarkable plant species, presumed to be a relict from the Pleistocene steppe vegetation, has a swivel root system, and it grows mostly on south-facing slopes with a pronounced summer drought period, and it generally enters into full dormancy in June [4]. 

At a subordinate taxonomic level in the hierarchy, the subspecies *Astragalus exscapus* ssp. *transsilvanicus* (Schur) Nyár. abbreviated *ASTRA*, is even lesser known in science. The subspecies was described by Podlech as a local endemism of the Transylvanian hills, a plant up to 40 cm tall, sparsely hairy, with a reduced stem and with lemon-yellow odoriferous flowers (Figure 1A) which appear in middle May, scattered in small populations [5]. The morphology of the root system of *A. exscapus* ssp. *transsilvanicus* (Figure 1B) is very similar to that of *A. membranaceus* (Fisch.) Bunge, a notorious medicinal plant, framed by the World Health Organization into the top 50 essential herbs used in the traditional Chinese medicine [6].

*A. membranaceus* roots, named Astragali Radix or Huang-Qi, revealed various remarkable biological functions, including potent immune-modulating, antioxidant, anti-inflammatory, and antitumor activities in modern pharmacological studies and clinical practices [7]. Therefore, there arises the hypothesis of analogous biological activities performed by the phytochemicals extracted from the roots of *A. exscapus* ssp. *transsilvanicus,* considering that several perennial legumes have been used as edible or medicinal plants since ancient times and they are characterized by high nutritional value, abundance of minerals and secondary metabolites [8].

According to Rocchetti et al. (2022), the interrelation between phenolic compounds and dietary fiber reveals certain health benefits promoted by noncovalent and covalent interactions [9]. It is presumed that these interactions are responsible for the carrier effect ascribed to fiber toward the digestive system, which can further modulate the bioaccessibility of phenolics, and therefore can shape health-promoting effects in vivo [9]. The primary constituents considered responsible for the therapeutic efficiency of the Huang-Qi extracts are phytochemicals from three major classes, namely polysaccharides, phenolics, and saponins. Focusing on the phenolic compounds, isoflavones belonging to flavonoids class are natural bioactive molecules consisting of a phenyl benzopyran skeleton formed by two phenyl rings joined by a heterocyclic pyran ring. They are commonly found in plant species, and possess good bioactivities which have been applied to enrich nutraceutical, medicinal, and cosmetic products [10]. 

Recent studies show that higher phenolic and flavonoid contents of *A. membranaceus* roots of different origins are in positive correlation with their antioxidant activities [11]. The study exemplifies many different subclasses of flavonoids described from Huang-Qi extracts including flavone, flavonols, flavanone, flavanonol, chalcone, aurone, isoflavone, and pterocarpan, highlighting that phenolics, precisely flavonoids, are considered accountable markers for quality evaluation and standardization of *A. membranaceus* and its processed products [11]. Among the isoflavones, formononetin, calycosin, and their 7-O-β-D-glucosides are abundant in Huang-Qi and, as part of the group of isoflavones, formononetin is highly relevant due to its biological properties, such as antioxidant, anticancer, and anti-inflammatory activities, together with others [12]. 

The aim of the research is to explore the nutritional composition and the bioactive constituents of the *ASTRA* roots, settling this way a basic foundation of knowledge for further complementary studies. Future perspectives of the research are linked to the biological activities (e.g., prebiotic properties and immune-modulating effects) and the possible applications in food science and nutrition.

From the best of our knowledge, the characterization of *Astragalus exscapus* ssp. *transsilvanicus* has been made from an ecological viewpoint [13] and no chemical profiling has been effectuated so far. The main objectives of the present study were to: (i) perform basic characterization of the *ASTRA* roots by proximate analysis, (ii) to investigate the fatty acid profile of the lipids extracted from the *ASTRA* roots, (iii) to examine the phenolic composition of the root extracts by LC-MS, and (iv) to evaluate the biological activities by determining the antioxidant, antimicrobial, and cytotoxic capacities of the extracts.

## 2. Results and Discussion 

### 2.1. Proximate Analysis 

The proximate analysis of the *ASTRA* roots was accomplished to find out the composition of macromolecules, more exactly the proportion of ash, protein, lipids, moisture, and carbohydrates levels. The results are presented in Table 1, showing that the predominant components of the *ASTRA* roots are carbohydrates with 82.868 ± 0.04%, and lipids with 8.471 ± 0.01%. Similar composition analysis regarding perennial legumes, as a reliable source of ingredients for healthy food, reported 46.2 and 43.5 g/100 g total carbohydrates for *Astragalus glycyphyllos* L. and *A. cicer* L., respectively [8]. The notable differences between the results might be given by the peculiar moisture content of the analyzed species or might be attributed to the geographical provenience, as well as to the specificity of the analyzed plant species.

For further analysis of the *ASTRA* roots, Fourier-transform infrared spectroscopy (FTIR) was used to obtain an infrared spectrum of absorption of the solid root sample, based on the interaction of electromagnetic radiation with the substance. The sample was exposed in a beam of continuous spectrum radiation in the IR range, obtaining an absorption profile (Figure 2).

Each substance has a unique IR spectrum, a spectral footprint on which it can be identified. The spectrum of the *ASTRA* root sample is dominated by an absorbance at 1018 cm^−1^ due to the total contribution of polysaccharides. At the same time, the spectrum is characterized by the typical absorptions of amide I and amide II protein bands, with an absorbance at 1620 and a shoulder at 1525 cm^−1^. The amide band I occurs due to the tensile vibration of the C=O bond and the amide band II is due to the deformation vibration C-N and C-NH.

The simple carbohydrates profile of the *ASTRA* roots was determined by HPLC coupled to a refractive index detector (RID) together with external standards (maltose, glucose and galactose, Cayman chemical company), to identify and quantify the individual carbohydrates content. The results showed a mean glucose content of 100.52 ± 0.69 mg/g DW followed by fructose with 37.21 ± 0.76 mg/g DW and maltose with 10.83 ± 0.18 mg/g DW. According to a study on the immune receptors for polysaccharides from the roots of *A. membranaceus*, the immunopotentiating effect has been associated with its polysaccharide fractions, and its composition analysis by gas chromatography presented a configuration of the molar ratios as Rha:Xyl:Glc:Gal:Man:Fru = 4.9:4.7:8.3:122.2:2.2:3.1 [14].

Furthermore a recent systematic review regarding phytochemistry, pharmacology, and pharmacokinetics on Astragali Radix, reports that over 30 polysaccharides have been isolated and are mainly classified into dextran and heteropolysaccharides [7]. In this regard further analysis of the *ASTRA* roots about the manifold polysaccharide composition are considered in perspective. 

### 2.2. Fatty Acid Composition Analysis of the ASTRA Roots by Gas Chromatography-Mass Spectrometry (GC-MS)

Given the results of the proximate analysis showing a moderate lipid content of the sample (8.471% DW), the investigation continued with gas chromatography-mass spectrometry analysis to identify and quantify the prevalent fatty acids from the *ASTRA* roots. 

The fatty acid profile, shown in Figure 3, presents four predominant compounds, namely linoleic acid C18:2 (n-6) as major compound reaching 31.10%, palmitic acid (C16:0) with 17.30%, oleic acid C18:1 (n-9) with 15.61%, and α-linolenic acid C18:3 (n-3) representing 14.21% out of the total fatty acids found in the samples.

Previous data linked to the fatty acid composition of some *Astragalus* species’ seeds from Turkey (e.g., *A. echinops* Aucher ex. Boiss., *A. subrobustos* Boriss., *A. jodostachys*, Boiss. and Buhse., *A. falcatus* Lam., and *A. fraxinifolius* DC.) contained linolenic acid ranging between 23 and 41%, followed by linoleic acid with 23 to 37%, and oleic acid in proportion of 8 to 19%, as most important components [15]. In the present study the fatty acid profile was determined for the lipids extracted from the *ASTRA* roots, and the results are represented in Table 2 showing linoleic acid as preponderant compound followed by palmitic, oleic, and α-linolenic acids. 

While linoleic acid is a polyunsaturated omega-6 (ω-6) fatty acid fundamental for human nutrition and the main source for its intake is through diet, α-linolenic acid is a polyunsaturated omega–3 (ω-3) essential fatty acid, which is needed for the formation of cell membranes and the synthesis of hormone-like compounds called eicosanoids (e.g., prostaglandins, thromboxanes, and leukotrienes), and which are significant regulators of blood pressure, blood clotting, and the immune response [16]. The ratio of (ω-6)/(ω-3) plays a fundamental role in the reduction in chronic diseases, especially of cardiovascular character [17], and furthermore, the optimal ratio between (ω-6) and (ω-3) was reported to be 2:1 to 3:1 [18]. The *ASTRA* root samples show a ratio of 2.19 between the (ω-6) and (ω-3) fatty acids, which is situated in the optimal range, and is comparable to canola oil and hempseed oil, highly recommended in vegetarian diets [19].

Along with the individual fatty acid composition, further calculations are presented in Table 2, with reference to certain quality indices. The sum of polyunsaturated fatty acids (PUFAs) reaches 45.31% while the sum of saturated fatty acids (SFAs) was 35.41% which further results in a ratio of 1.28 PUFAs/SFAs in the analyzed samples. The ratio of saturated to unsaturated fatty acids constitutes an important property of the phospholipid composition of biological membranes. Changes to this proportion are thought to exert deleterious effects on cells and, in particular, reduction in the number of unsaturated fatty acids in membranes may contribute to the development of a number of pathophysiological states, such as cardiovascular disease, diabetes, and cancer [20]. According to a recent mini-review about nutritional indices, the ratio of saturated to unsaturated fatty acids is normally used to assess the impact of diet on cardiovascular health and it hypothesizes that PUFAs in the diet can depress low-density lipoprotein cholesterol and lower levels of serum cholesterol, whereas the SFAs contribute to high levels of serum cholesterol [21]. Accordingly, the higher this ratio, the more positive the effect on human health. The *ASTRA* roots have a PUFAs/SFAs ratio comparable to shellfish *Pyura chilensis* (1.31), fish *Lagocephalus guentheri* 1.3 or brown seaweed *Spatoglossum asperum* 1.38 according to the fatty acid evaluation reported earlier [21].

Among the identified monounsaturated fatty acids palmitoleic acid C16:1 (n-7) and its isomer cis-7-hexadecenoic acid C16:1 (n-9) was quantified in 0.34 and 0.89% from the total fatty acid composition, respectively. Generally, monounsaturated fatty acids are “good” to biological membranes because, being liquid at body temperature yet not easily oxidized, they help to maintain membrane fluidity within the appropriate limits. Previous studies showed the role of palmitoleic acid as a lipid hormone that coordinates cross-talk between liver and adipose tissue and exerts anti-inflammatory protective effects on hepatic steatosis in animal models. Furthermore, 16:1n-7 and 16:1n-9 manifest strong anti-inflammatory activity when added to phagocytic cells at low concentrations and therefore palmitoleic acid might be a promising anti-inflammatory lipid that may help to ameliorate metabolic disorders [20]. 

### 2.3. Qualitateive and Quantitative Analysys of the Extracts by LC-DAD-ESI-MS

The *ASTRA* root powder was subjected to ultrasound assisted extraction using three different solvents (methanol 50%, ethanol 50% and water) in order to examine how the solvent influences the individual composition of the extracts, and to select the most efficient one for subsequent biological activity tests. 

The obtained extracts were analyzed by LC-DAD-ESI-MS, and the parameters of the seven identified compounds in the *ASTRA* root extracts are presented in Table 3. Reference standards (calycosin, formononetin, calycosin-7-O-β-D-glucoside, formononetin-7-O-β-D-glucoside, and Cayman chemical company) were used to identify the individual isoflavones. Calycosin-7-O-β-D-glucoside-6-O-malonate and ononin were identified based on their molecular weight and UV spectra, as reported previously by Jing-Zheng Song [22]. 

The LC results are summarized in Table 4 and the values are expressed as µg/g DW ± standard deviation. Consistent differences can be observed in the composition of the extracts regarding isoflavones content; however, the prevalent bioactive compound identified in each extract was Astragaloside IV with values ranging between 389.13 µg/g DW and 425.32 µg/g DW in the ethanol extract and the methanol extract, respectively.

One of the major bioactive compounds found in *A. membranaceus* root is Astragaloside IV, a triterpenoid saponine unique to the *Astragalus* species. Astragaloside IV demonstrated potential cardio-protective and immunological enhancement activities through experimentation in vitro and in vivo, illustrated by multiple pharmacologic effects, including anti-inflammatory, antifibrotic, antioxidative stress, anti-asthma, antidiabetes, immunoregulation, and cardioprotective effect via numerous signaling pathways [23]. 

In the present study Astragaloside IV, was identified in each extract obtained from the *ASTRA* roots and quantified as well based on reference standard. A recent systematic review regarding the neuroprotective effect in experimental models of neurological disorders shows that the administration of Astragaloside IV can improve behavioral and neurochemical deficits largely due to its antioxidant, antiapoptotic, and anti-inflammatory properties, emerging as an alternative therapeutic approach for the treatment of neurological disorders [24]. In the present study the highest quantity of Astragaloside IV was found in the methanol extract, followed by the ethanol extract; nevertheless, is important to mention that the water extract contained 391.70 µg/g DW, which is an insignificantly slight difference between the results, considering the cost efficiency of the used solvent, and therefore water extraction assisted by ultrasound is an appropriate method to obtain Astragaloside IV-rich extracts with minimal expenses.

Eight flavonoids were identified in *A. membranaceus* roots by UPLC-electrospray ionization mass spectrometry, among them calycosin-7-O-β-D-glucoside, calycosin-7-O-b-D-glucoside-6′-O-malonate, ononin, formononetin-7-O-β-D-glucoside-6′-O-malonate, as well as formononetin [25]. All the above-mentioned compounds were present in variable amounts in each *ASTRA* root extract, indicating a solvent dependent affinity of certain molecules. The quantification of isoflavones was made based on the calibration curve of formononetin standard (y = 126.44x + 653.3).

The quantity of Calycosin-glucoside-malonate decreased in the ethanol extract nearly two-fold, while in the methanol and water extract it presented 38.81 ± 0.26 µg formononetin/g DW and 37.07 ± 0.13 µg formononetin/g DW, respectively. In a similar way, Formononetin-glucoside-malonate presented high values in the methanol and ethanol extracts, with values of 91.76 ± 0.10 µg formononetin/g DW and 90.85 ± 0.13 µg formononetin/g DW, respectively, while the water extract showed a consistent decrease in the formononetin-glucoside-malonate content (26.71 ± 0.06 µg formononetin/g DW).

Calycosin-glucoside presented a good compatibility with ethanol, showing 27.89 ± 0.13 µg formononetin/g DW, a moderate affinity with methanol showing 15.91 ± 0.16 µg formononetin/g DW and a considerably smaller content was observed in the water extract with an average value of 2.51 ± 0.32 µg formononetin/g DW. According to Guo et al. calycosin-7-O-β-D-glucoside has been proved to exhibit several pharmacological activities, such as anti-inflammatory, antioxidative, and neuroprotective activities [7,26]; therefore, further investigations are desirable to elucidate its health benefits and mechanism of action. 

Calycosin-glucoside-malonate was identified and quantified as well in the *ASTRA* root extracts with values ranging between 20.38 and 38.81 µg formononetin/g in the ethanol and methanol extract, respectively, being comparable with earlier findings. Previous studies performed on the simultaneous determination of isoflavonoids and saponins in Astragali Radix showed values ranging between 26.44 and 237.48 mg/100 g and the differences between samples might be deriving from several factors such as provenience, the plants age, and extraction conditions [22].

Formononetin present in the water extract in highest amounts 17.77 ± 0.13 µg/g DW, is a methoxylated isoflavone and it is considered the vital bioactive compound of *A. membranaceus*, having various pharmacological activities [7]. With all that calycosin-glucoside, calycosin and formononetin presented considerably higher values in the ethanol extract, the sum of phenolic compounds was most consistent in the methenol extract with 578.33 ± 0.42 µg/g DW and further biological activity tests were conducted on the basis of the methanol extract.

Specifically, isoflavones are a group of phytoestrogens of relevant interest in the areas of nutrition, medicine, and cosmetics; however, it is still difficult to state their general health effects, since isoflavones are constituted by a great diversity of phytoestrogens, and each one may have different activities, pharmaco-kinetic properties, and/or metabolic pathways. A recent comprehensive review paper with regards to the biological effects and uses of formononetin highlights the increasing interest in exploring its potential applications in food and human nutrition due to the performance aspects of formononetin and its mechanism of action related to menopause, neurodegenerative diseases, and metabolic syndrome [12]. 

The high incidence of non-communicable diseases and autoimmune diseases as well as metabolic disorders has increased the awareness of consumers, who are looking for “healthy” food. Beyond nutrients, the diet provides a vast source of bioactive compounds able to modulate several biological actions in living organisms, cells or tissues; however, there are distinctive and particular factors in each individual, such as enterohepatic enzyme activities and intestinal microbiota, which will significantly contribute to the pharmacokinetic properties of the bioactive compounds as well [12]. These specific aspects might alter the absorption and metabolism of the bioactive compounds, resulting in different biological properties when ingested. A recent overview related to the potential benefits of incorporating *A. membranaceus* into the diet of people undergoing disease treatment summarized the positive effects on general health state and reduced risk of neurodegenerative diseases, type 1 and 2 diabetes and cancers development, documented by a vast number of studies [27]. In perspective, the concept of personalized nutrition might be suitable for disease prevention and/or treatment, which might further contribute to a sustainable health in an increasingly aging population [28]; therefore, the integration of alternative edible or medicinal plant sources into functional food products, and the screening of local endemic plant species for this purpose, is an evergreen aim of the scientific community at a multidisciplinary level.

### 2.4. Bioactive Composition and Antioxidant Activity Analysis

The evaluation of the bioactive composition and the antioxidant activity of the *ASTRA* root was performed on the methanol extract, obtained by using a 1:10 (*w*/*v*) ratio of biological material to solvent, beyond any other concentration processes. The results are summarized in Table 5. 

Phenolic compounds are generally recognized to be accountable for biological activities such as antioxidant and antiaging effects, as well as other beneficial actions [11]. Yet, the biological activities cannot be attributed to a single class of compounds but more to the synergistic effect of several phytochemicals (e.g., saponins, polysaccharides and/or phenolic compounds). 

The bioactive composition revealed by the total phenolic content of the *ASTRA* root extracts indicates 110.79 µg GAE/mL and the total flavonoids content showed 14.81 ± 2.22 µg Quercetin/mL extract. Considering the 1:10 (*w*/*v*) concentration of the tested extracts as well as the multiple results reported previously on manifold *Astragalus* species, together with distinct geographical origins and several extraction methods applied, it would be difficult to make an objective comparison in between the results. 

DPPH has been broadly used as a method for determining the free radical scavenging capacity of extracts. In the present investigation, DPPH radical scavenging activity showed a value of 463.51 ± 2.59 µM Trolox/mL extract. Previous results regarding antioxidant activity of methanolic extracts from different parts of the *Astragalus ponticus* Pall plant ranged between 8.86 ± 0.92 and 43.64 ± 0.94 mg TEs/g extract, showing highest values for the leaves extract followed by the root extracts [29].

Former studies conducted on *Astragalus* species investigated the free radical scavenging efficiency through multiple methods (e.g., FRAP, ABTS, ORAC, CUPRAC), highlighting the unexplored association between the antioxidant activity and the total phenolics/flavonoids content, and the necessity to further enlighten the synergistic effect of multiple bioactive compounds on enhanced biological activities [11,12,24,29,30].

### 2.5. Antimicrobial Activity Assay

The antimicrobial effects of the *ASTRA* root extracts (methanol, ethanol and water extracts) were tested first on four bacterial strains (*S. aureus*, *E. coli*, *P. aeruginosa*, *S. typhimurium*) and two yeasts (*C. ablicans*, *C. parapsylosis*) by determining the minimum inhibitory concentration (MIC). The extracts, however, were obtained by using a ratio of 1:40 (*w*/*v*) *ASTRA* root powder to solvent, and the preliminary results showed a moderate antimicrobial effect of the methanol extract on *S. aureus*. Therefore, the investigation continued with two additional bacterial strains, namely *S. epidermidis* and *S. enterica* and with a methanol extract which was concentrated under vacuum. The final concentrations of the tested extracts were 0.360 g dry weight (DW)/mL for sample A1, 0.380 g DW/mL for sample A2 and 0.365 g DW/mL for sample A3, respectively. The results are expressed as mg dry weight (DW)/mL extract inhibiting the growth of bacterial strains, and are presented in Table 6.

Significant results were observed on *S. aureus* and *S. epidermidis* bacterial strains as their growth was inhibited by the methanol extract with values situated between 0.356 and 0.703 mg *ASTRA* root powder (DW)/mL. 

*S. epidermidis* is mostly involved with indwelling medical device-associated infections, and together with its more virulent relative *S. aureus*, ranks first among the causative agents of nosocomial infections according to Michael Otto [31]. Although *S. epidermidis* infections only rarely develop into life-threatening diseases, their frequency, and the fact that they are extremely difficult to treat, represents a serious burden for the public health system. The costs related to vascular catheter-related bloodstream infections caused by *S. epidermidis* was estimated to 2000 million USD annually in the United States alone as the treatment is complicated by the presence of specific antibiotic resistance genes and the formation of biofilms [31]. 

Previous studies conducted on the antimicrobial effects of several *Astragalus* species showed potential to inhibit some Gram-positive and Gram-negative bacterial strains such as *Staphylococcus aureus*, *Staphylococcus epidermidis*, *Enterococcus faecalis Escherichia coli*, *Klebsiella pneumonia*, *Pseudomonas aeruginosa*, and *Proteus vulgaris* at MIC values ranging between 5.20 to 40 mg/mL [8,32,33,34]. 

The results of the present antimicrobial activity assay, show remarkable antimicrobial potential of the *ASTRA* roots at a very low concentration, more exactly at 0.356 and 0.703 mg *ASTRA* root powder (DW)/mL highlighting its efficacy to inhibit *S. aureus* and *S. epidermidis* bacterial strains.

### 2.6. Cell Proliferation Assay

To carry out the in vitro experiments, two skin cell lines were chosen for the study: Hs 27, normal human fibroblasts, and B16F10, murine melanoma. The antiproliferative potential of the treatment on both cell lines was evaluated and the results are shown in Figure 4. 

In the case of the B16F10 tumor cell line, the treatment had a strong inhibitory effect in a concentration-dependent manner. The IC50 value was identified at the concentration of 26.4 μg/mL. The lowest concentration, 5 μg/mL, had a downwards effect on cell proliferation, with no significant differences comparing to the control. However, as the dose increased to concentrations of 50 μg/mL, the cell viability decreased by up to about 65%. 

By contrast, in the case of the normal cell line Hs 27, the treatment applied under the same conditions and concentrations had no effect on the cells, even at the concentration of 50 μg/mL having an insignificant effect compared to the control. 

These results are also supported by the data obtained following the analysis of LDH, an indicator of cellular degradation. According to the graph in Figure 5, with increasing treatment concentration, the amount of LDH released underwent a significant increase, which demonstrates the disintegration of the membrane of a large number of cells and the release of LDH from the cytoplasm into the cell culture medium.

Similar to the viability assay, in the case of the Hs 27 cell line, the treatment did not result in significant cell degradation. In conclusion of these studies, the remarkable efficiency of the treatment in inhibiting the proliferation of B16F10 melanoma cells can be emphasized in a dose dependent manner. Previous studies of *A. membranaceus* linked to cell proliferation, growth inhibition and/or apoptosis on different cell lines (e.g., HT-29 human colon cancer cells, BGC-823 gastric cancer cells, myeloma, macrophages, human tumor cell line, THP-1) indicate good results, nevertheless the positive effects of the treatments are attributed to different class of compounds such as saponins, polysaccharides or flavonoids [7,12,14,35,36,37], further indicating the necessity of subsequent investigations to unfold the complexity of the subject. 

## 3. Materials and Methods

### 3.1. Chemicals and Solvents 

All the chemicals and standards used in the present study were of analytical grade. Methanol, ethanol, acetonitrile, acetic acid, DPPH, Trolox, sodium carbonate, aluminum chloride, resazurin sodium salt, gallic acid, Folin-Ciocalteu’s phenol reagent were purchased from Sigma-Aldrich (Steinheim, Germany) as well as the fatty acid standards, and the other chemicals implied in the oil extraction, fractionation, and preparation of fatty acid methyl esters (FAMEs). External standards such as calycosin-7-O-beta-D-glucoside, formononetin-7-O-beta-D-glucoside, calycosin, formononetin and astragaloside IV, were purchased from Cayman chemical company, as well as maltose, glucose and galactose, purchased from Merck—Germany. The chemicals used for the antimicrobial assays, such as triptic soy agar, Muller–Hinton agar and nutrient broth, were purchased from Himedia laboratories (Einhausen, Germany).

### 3.2. Plant Material and Proximate Analysis 

*ASTRA* roots were collected from Cojocna, (Beleni hill), Cluj County, Romania in April 2021 and no specific permission was required for sampling this location. The roots were cleaned, dried, grounded into a fine powder and maintained in sealed containers at room temperature until further analysis. 

The dry weight of the *ASTRA* roots was calculated according to the official method 930.15 (AOAC, 2005), and expressed in percentage. 

Ash was determined by carbon removal of 1 g powdered *ASTRA* root, which was incinerated in a muffle furnace at 550 °C for 4 h, in triplicate. The crucibles were removed from the heat to cool, and the weight loss was calculated. The total ash was expressed as a percentage of the dry weight. 

The protein content was investigated by Kjeldahl method and the total protein content was determined by using the general factor (6.25) as previously described [38]. Data were expressed as percent of dry weight. 

Lipids were extracted from *ASTRA* root sample with hexane using a Soxhlet extraction apparatus for 8 h. The solvent was removed by a rotary evaporator at 40 °C, and the residue was weighed and stored at −20 °C until further GC-MS analysis. The results were expressed as the lipid percentage in the dry *ASTRA* roots.

Carbohydrate content was estimated by difference of mean values: 100−(sum of percentages of moisture, ash, protein, and lipids) [38], and the solid root was examined by FTIR to obtain infrared spectrum of absorption, and to detect different functional groups in the ASTRA sample.

### 3.3. Identification of the Monosaccharides Found in the ASTRA Roots by HPLC–RID

The simple sugar content (maltose, glucose, fructose) of the *ASTRA* roots was determined by HPLC coupled to a refractive index detector (RID). Briefly 0.5 g of sample was extracted with 10 mL of double distilled water in a sonication unit (Elmasonic E 15 H) for 60 min at 65 °C, followed by centrifugation at 10,000 rpm for 10 min at 24 °C. The supernatant was filtered through Chromafil Xtra PA-45/13 0.45 µm nylon filter and 20 µL extract was injected into the HPLC system (Agilent 1200 HPLC Agilent Technologies, Santa Clara, CA, USA) Compound separation was performed on a Polaris Hi-Plex H column, 300 × 7.7 mm (Agilent Technologies, CA, USA), using mobile phase H_2_SO_4_ 5mM at a flow rate of 0.6 mL/min, column temperature of 70 °C and RID temperature of 35 °C. 

Data acquisition and interpretation of the results were performed using OpenLab-ChemStation software (Agilent Technologies, CA, USA). Identification of carbohydrates in the analyzed samples was made by comparing the retention time of the peak in the sample with that of standard carbohydrates. For the quantification of simple sugars in the samples, calibration curves were performed with standard substances of purity >99%, by injecting 5 different concentrations in the range 1–10 mg/mL.

### 3.4. Fatty Acid Analysis by Gas Chromatography-Mass Spectrometry (GC-MS)

For the investigation of the fatty acid profile of the *ASTRA* roots, an acid-catalyzed procedure was used. Briefly, a sample of the total lipid extract (10–15 mg) was transesterified into fatty acid methyl esters (FAMEs), as indicated in a previous study [39]. The apparatus used for this analysis was a GC-MS (PerkinElmer Clarus 600 T GC–MS, PerkinElmer, Inc., Shelton, CT, USA) equipped with a Supelcowax 10 capillary column (60 m × 0.25 mm i.d., 0.25 μm film thickness; Supelco Inc., Bellefonte, PA, USA). The temperature in the column was programmed to increase with a rate of 7 °C/min, from 140 up to 220 °C and maintained there for 23 min. The utilized carrier gas was helium, with a continuous flow rate of 0.8 mL/min. 

The FAMEs were identified by comparing their retention times to those of the commercial standard mix of fatty acids (37 components FAME Mix, Supelco no. 47885-U) and the resulted mass spectra to those found in our database (NIST MS Search 2.0). The same database of compounds was used to quantify each fatty acid, by calculating the individual peak area percentage from the total amount of fatty acids.

### 3.5. Ultrasound Assisted Extraction of Isoflavones

Three trials of extraction were experimented with water, methanol in water (1:1 *v*/*v*), and ethanol in water (1:1 *v*/*v*) to examine the individual isoflavones from each extract. The ratio of sample to solvent was 1:40 (*w*/*v*) according to a previous study conducted on isoflavonoids extraction assisted by ultrasound [40]. Briefly, the powder sample (1 g) was added with 40 mL of solvent, homogenized, and placed into an ultrasonic bath (Elma Schmidbauer GmbH, Singen, Germany) for 20 min at room temperature, followed by centrifugation at 10.8× *g* for 10 min. The supernatant was collected, filtered, and injected to the HPLC system.

### 3.6. Qualitative and Quantitative Determinations of Isoflavones

An Agilent 1200 HPLC equipped with a diode array detector (DAD), coupled to an MS-detector single-quadrupole Agilent 6110 (Agilent Technologies, Santa Clara, CA, USA) was used to identify individual isoflavones. The separation of the compounds was performed at 25 °C on an Kinetex XB C18 (4.6 × 150 mm,5 µm) column. The binary gradient consisted of 0.1% acetic acid in distilled water (99:1 *v*/*v*) as solvent A, and 0.1% acetic acid in acetonitrile (*v*/*v*) as solvent B at a flow rate of 0.5 mL/min.

Various isoflavones were identified by comparing the retention times, UV–visible and mass spectra of unknown peaks with spectra of the standards. For MS fragmentation, the ESI (+) module was applied, with a scanning range between 120 and 1200 *m*/*z*, capillary voltage 3000 V, at 350 °C and with a nitrogen flow of 7 L/min. The phenolic compounds were monitored by DAD, and the absorption spectra (200–600 nm) were collected continuously during each run. Data analysis was performed using Agilent ChemStation Software (Rev B.04.02 SP1, Palo Alto, CA, USA).

### 3.7. Bioactive Composition and Antioxidant Activity Analysis

#### 3.7.1. Total Phenolic Content 

The total phenolic content was determined by the Folin–Ciocalteu reagent method using the methanol extract obtained by UAE of *ASTRA* roots. The absorbance of the extract was measured at 760 nm using a spectrophotometer (Perkin Elmer Precisely, Lambda 25, Waltham, MA, USA) and the obtained values were interpreted using a standard curve expressed by gallic acid.

#### 3.7.2. Total Flavonoids Content

The AlCl_3_ colorimetric method was used to determine the total flavonoid content of the methanol extract obtained by UAE of *ASTRA* roots. The absorbance was measured at 510 nm and then converted to content through a standard curve expressed by quercetin.

#### 3.7.3. Antioxidant Activity by DPPH

The antioxidant activity of the *ASTRA* root extracts was tested using the DPPH (1,1-diphenyl-2-picrylhydrazyl) free scavenging capacity method of Cuvelier, and Berset [41], as described previously by Dulf et al. [39,42]. The findings were reported as micromolar Trolox equivalents (μmol TE)/100 g sample (DW of *ASTRA* roots powder).

### 3.8. Antimicrobial Capacities

The minimum inhibitory concentration (MIC) was determined for the following standard strains: *Escherichia coli* ATCC 25922, *Staphylococcus aureus* ATCC 29213, *Staphylococcus epidermis* ATCC 12228, *Pseudomonas aeruginosa* ATCC 27,853 and *Salmonella enterica* NCTC 6017. The microorganisms were grown on specific agar media in the Food Biotechnology Laboratory of the University of Agricultural Sciences and Veterinary Medicine Cluj-Napoca, Romania. Tryptic Soy agar (Himedia, M1968) was used for *E. coli* and *P. aeruginosa*, and Mueller–Hinton agar (Oxoid Ltd., Basingstoke, Hampshire, UK) for *S. epidermidis*, *S. aureus*, and *S. enterica*. Plates were incubated for 24 h at 37 °C and the bacterial morphology was confirmed by optical microscopy.

Further, the methanol extracts were subjected to MIC determination, as described by Vodnar et al. (2017) [43]. First, 100 µL of sterile nutrient broth (Oxoid Ltd., Basingstoke, Hampshire, UK) were added in the wells of a 96-well microplate. Then, 100 µL of extract was added in the first well, followed by serial 11-fold dilutions, discarding the surplus of the last well of each row. Further, the assay continued with the addition of 10 µL of inoculum (1.5 × 10^5^ CFU/mL) and the incubation of the microplates for 20–22 h at 37 °C for bacterial growth and the MIC of the samples was determined after the addition of 20 μL (0.2 mg/mL) of resazurin solution, followed by incubation for another 2 h at 37 °C. Color change from blue to pink indicated the reduction in resazurin, thus the presence of viable bacteria. The MIC was determined as the lowest concentration of the *ASTRA* root extracts (µg extract/mL) inhibiting the visual growth of the test culture on the microplate.

### 3.9. Cell Culture 

B16-F10 murine melanoma and Hs27 human fibroblast cells purchased from American Type Culture Collection (ATCC) were maintained in DMEM media, supplemented with FBS (10%), 1 mM glutamine, 1% antibiotics, in standard conditions.

#### 3.9.1. Cell Proliferation Assay

Cell proliferation was measured by MTT assay. Briefly, both cell lines, 8 × 10^3^ cells/well were seeded in 96-well plates for 24 h. further, the media was replaced by fresh one containing the treatment, in concentrations ranging from 0 to 50 μg/mL, for another 24 h. The media from each experimental well was then replaced with MTT reagent solution (0.5 mg/mL), discarded after 2 h and the formazan crystals formed were dissolved in DMSO. The absorbance of solubilized formazan formed in viable cells was measured at 550 and 630 nm (for sample and background, respectively) using the microplate reader (HT BioTek Synergy, BioTek Instruments, Winooski, VT, USA). The results were expressed as percent survival relative to the untreated control. Each treatment was repeated three times and each repetition had five experimental wells for each concentration.

#### 3.9.2. Measurement of LDH Release Level

LDH release assay was performed according to kit instructions (Pierce LDH Cytotoxicity Assay Kit, Rockford, IL, USA) and the cells were treated as described for the proliferation assay. The absorbance at 490 nm and 680 nm, respectively, was measured using the microplate reader (BioTek Instruments, Winooski, VT, USA) and the results were expressed as percentage variation of LDH release normalized to control.

## 4. Conclusions

The primary compounds found in the *ASTRA* roots were polysaccharides and lipids, and the fatty acid composition showed linoleic acid as preponderant compound, followed by palmitic, oleic and α-linolenic acids. The methanol extract of the *ASTRA* roots presented highest phenolic content, Astragaloside IV being the predominant compound. The antimicrobial assay showed remarkable antimicrobial potential of the extract, highlighting its efficacy to inhibit *S. aureus* and *S. epidermidis* bacterial strains. Likewise, the noteworthy proficiency of the treatment in inhibiting the proliferation of B16F10 melanoma cells can be emphasized. Overall, the present study represents a humble introduction to the manifold inquiry regarding *ASTRA* roots’ biological activities, potential health benefits and its applicability in nutraceutical, pharmaceutical and/or functional food products. 

## Figures and Tables

**Figure 1 ijms-23-15161-f001:**
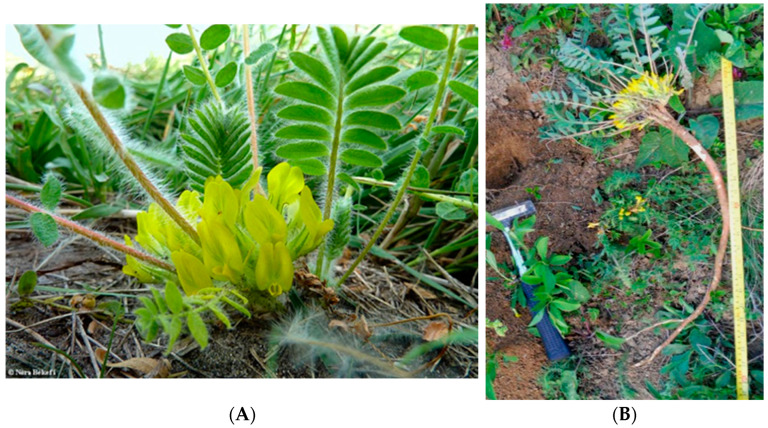
*Astragalus exscapus* ssp. *transsilvanicus* (**A**) aerial part (**B**) root system. Source: original photo.

**Figure 2 ijms-23-15161-f002:**
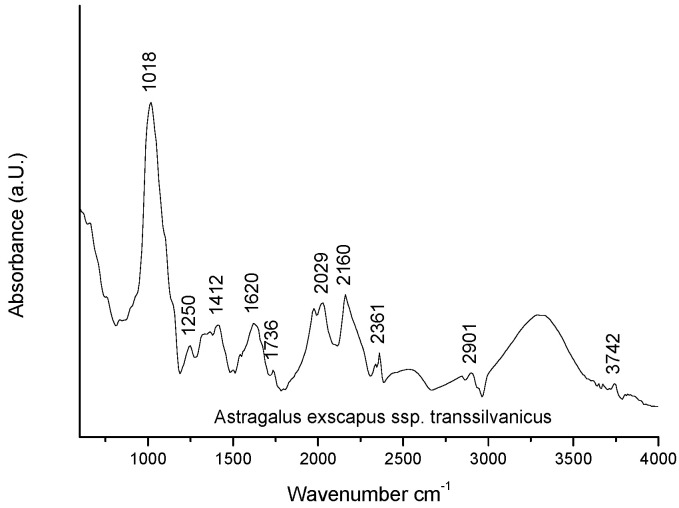
Fourier Transform Infrared spectroscopy profile of *ASTRA* roots.

**Figure 3 ijms-23-15161-f003:**
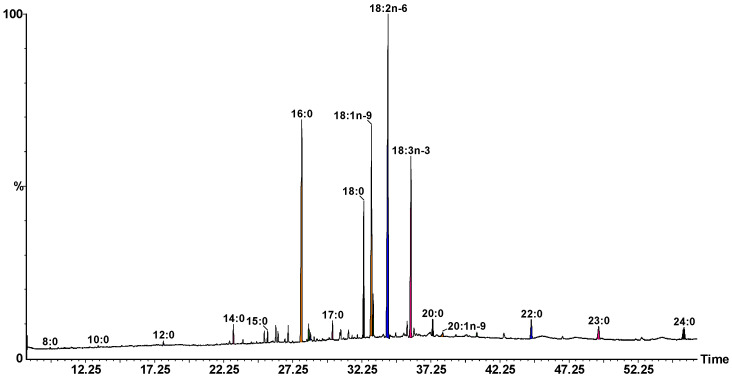
Fatty acid profile of *ASTRA* roots determined by GC-MS.

**Figure 4 ijms-23-15161-f004:**
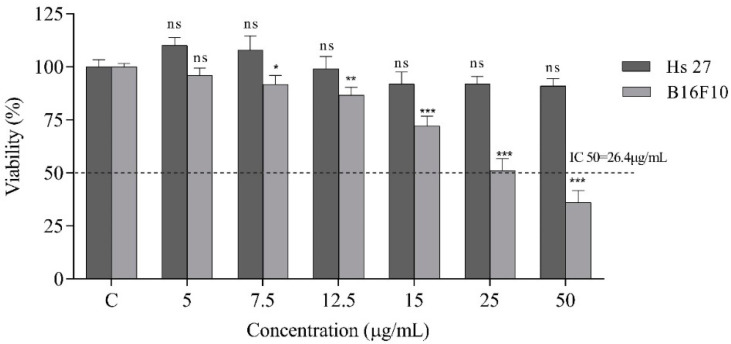
Antiproliferative effect of treatment by different concentrations (0–50 µg/mL) on HS-27 (normal human fibroblasts) and B16F10 (murine melanoma cells) after 24 h. Data represent the means ± SEM of at least three independent experiments (significant differences, * *p* < 0.05, ** *p* < 0.01, *** *p* < 0.001, n.s. no significant differences).

**Figure 5 ijms-23-15161-f005:**
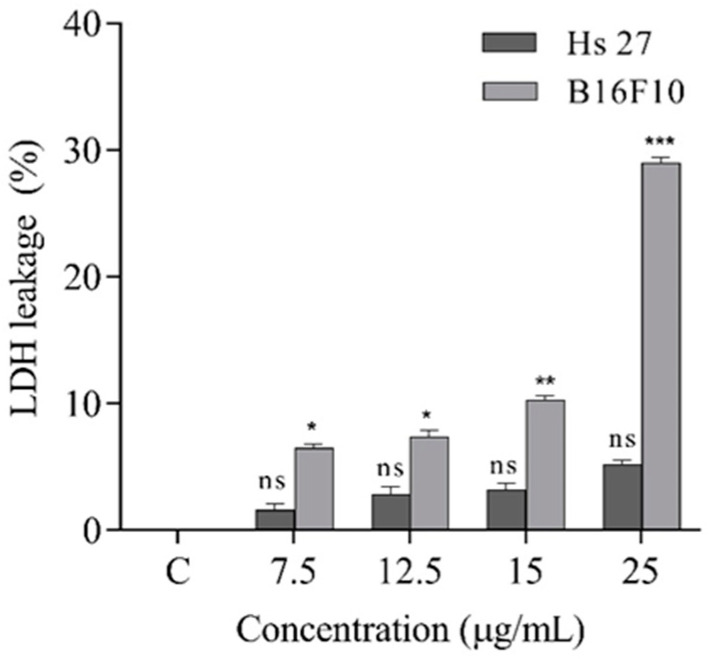
Effects of treatment on the lactate dehydrogenase (LDH) release in both cell lines after 24 h of exposure to treatment. Data represent the means ± SEM of at least three independent experiments (significant differences, * *p* < 0.05, ** *p* < 0.01, *** *p* < 0.001, n.s. no significant differences).

**Table 1 ijms-23-15161-t001:** Proximate analysis of the *ASTRA* roots.

	%
Ash (DW)	0.021 ± 0.01
Protein (DW)	1.042 ± 0.02
Lipids (DW)	8.471 ± 0.01
Moisture	6.598 ± 0.06
Carbohydrates (DW)	82.868 ± 0.04

**Table 2 ijms-23-15161-t002:** Fatty acid composition of the *ASTRA* roots.

Identified Fatty Acids	%
C 8:0 Caprylic acid	0.06
C 10:0 Capric acid	0.07
C 12:0 Lauric acid	0.19
C 14:0 Myrsitic acid	0.93
C 15:0 Pentadecanoic acid	0.54
C 16:0 Palmitic acid	17.30
C 16:1 (n-9) cis-7-hexadecenoic acid	0.89
C 16:1 (n-7) palmitoleic acid	0.34
C 17:0 Margaric acid	0.88
C 18:0 Stearic acid	8.76
C 18:1 (n-9) Oleic acid	15.61
C 18:1 (n-7) Vaccenic acid	1.95
C 18:2 n-6 linoleic acid	31.10
C 18:3 n-3 α-linolenic acid	14.21
C 20:0 Arachidic acid	1.10
C 20:1 n-9 Eicosenoic acid	0.50
C 21:0 Heneicosylic acid	0.32
C 22:0 Behenic acid	1.94
C 23:0 Tricosylic acid	1.54
C 24:0 Lignoceric acid	1.77
Total	100.00
Ʃ SFAs	35.41
Ʃ MUFAs	19.28
Ʃ PUFAs	45.30
Ʃ n-3 PUFAs	14.21
Ʃ n-6 PUFAs	31.10
n-6/n-3	2.19
PUFAs/SFAs	1.28

Sum of the saturated fatty acids-Ʃ SFAs; sum of monounsaturated fatty acids-Ʃ MUFAs; sum of polyunsaturated fatty acids-Ʃ PUFAs; sum of ω-3 polyunsaturated fatty acids-Ʃ n-3 PUFAs; sum of ω-6 polyunsaturated fatty acids-Ʃn-6 PUFAs; ratio of ω-6 and ω-3 fatty acids-n-6/n-3; ratio of saturated to unsaturated fatty acids-PUFAs/SFAs.

**Table 3 ijms-23-15161-t003:** Retention time (Rt), [M+H]^+^ and other MS ions, UV **λ**_max_ of the isoflavones identified from the *ASTRA* root extracts.

Peak No.	Retention Time R_t_ (min)	UV λ_max_ (nm)	[M+H]^+^ (m/z)	Compound	Subclass
1	12.53	257, 286 (sh)	533	Calycosin-glucoside-malonate	Isoflavone
2	15.64	257, 286 (sh)	447	Calycosin-glucoside	Isoflavone
3	17.56	250, 300 (sh)	517	Formononetin-glucoside-malonate	Isoflavone
4	19.11	250, 300 (sh)	431	Formononetin-glucoside (Ononin)	Isoflavone
5	20.85	257, 286 (sh)	285	Calycosin	Isoflavone
6	23.61	250, 300 (sh)	269	Formononetin	Isoflavone
7	25.85	245	785	Astragaloside IV	Triterpene

**Table 4 ijms-23-15161-t004:** Qualitative and quantitative analysis of the phenolic compounds found in the *ASTRA* root extracts by liquid chromatography expressed as µg formononetin/g DW.

Peak No.	Compound	Methanol Extract (50%)	Etanol Extract (50%)	Water Extract
1	Calycosin-glucoside-malonate	38.81 ± 0.26	20.38 ± 0.06	37.07 ± 0.13
2	Calycosin-glucoside	15.91 ± 0.16	27.89 ± 0.13	2.51 ± 0.32
3	Formononetin-glucoside-malonate	91.76± 0.10	90.85 ±0.13	26.71 ± 0.06
4	Formononetin-glucoside (Ononin)	1.99 ± 0.03	1.20 ±0.03	0.69 ±0.13
5	Calycosin	2.11 ± 0.10	16.90 ± 0.19	5.47 ± 0.16
6	Formononetin	2.43 ± 0.39	12.67 ± 0.10	17.77 ± 0.13
7	Astragaloside IV	425.32 ± 0.06	389.13 ± 0.10	391.70 ± 0.39
	Sum of phenolic compounds	578.33 ± 0.42	559.03 ± 0.25	481.92 ± 0.10

**Table 5 ijms-23-15161-t005:** Total phenolic content, total flavonoids content, and antioxidant activity by DPPH method of *ASTRA* root extract.

Biological Activities	
TPC (µg GAE/mL)	110.79 ± 4.00
TFC (µg QE/mL)	14.81 ± 2.22
DPPH (µM Trolox)	463.51 ± 2.59

**Table 6 ijms-23-15161-t006:** Minimum inhibitory concentration of the methanol extract on Gram-positive and Gram-negative bacterial strains expressed as mg *ASTRA* root powder (DW)/mL.

	Gram (+) Bacteria			Gram (−) Bacteria	
Sample	*S. aureus*	*S. epidermis*	*E. coli*	*P. aeruginosa*	*S. enterica*
A1	0.703	0.703	n.b.	n.b.	n.b.
A2	0.371	0.371	n.b.	n.b.	n.b.
A3	0.356	0.356	n.b.	n.b.	n.b.

n.b. no bioactivity.

## Data Availability

Not applicable.

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
