# Peer review of "Evaluation of the Astragalus exscapus L. subsp. transsilvanicus Roots’ Chemical Profile, Phenolic Composition and Biological Activities"

_ijms, 2022, doi:10.3390/ijms232315161_

Round 1
Reviewer 1 Report
the authors have already published a work in Plants on Astragalus exscapus, so they know the subject.
In my opinion, introduction should be improved, describing even more in detail what are the aims of the research and not just mentioning them. Furthermore, the presentation of the experimental design is a bit confusing, because at first it refers to some techniques, which are not always found in experimental practice.
The Authors report that they have identified lipids, fatty acids and carbohydrates present in the extract using the GC-MS technique, but do not specify whether these data have been compared with data reported in the literature, which may be comparative (e.g. data reported in the NIST). I'd be curious to see the mass spectroscopy data.
Moreover, when the IR spectrum of the extract is described, it makes particular reference to the streaching of the amide, but I cannot understand the reason (can many of the compounds present in the extract have this type of bond? if so, which ones? why are they the most abundant in the extract?)
Regarding the identification of carbohydrates, the Authors report the use of HPLC with the comparison with the reference standard: is it possible to see the chromatograms? Why wasn't a mass spectroscopy analysis done in this case, too?
The authors speak of extraction made with three types of solvents (50% ethanol, water and 50% methanol): have you tried other percentages for organic solvents? if so, with what results? Furthermore, the Authors underline that the choice of solvent changes, albeit slightly, the presence of some extracted substances: how could this change the response on the biological tests performed?
Moreover, it would be interesting to know in which period of the year the root was harvested and how the content of the substances present in it can vary according to the season.
Author Response
We would like to kindly express our appreciation for your time and effort in reviewing the manuscript. The suggested modifications were helpful to improve the quality and the content of the paper, therefore we agreed and applied the comments and recommendations as described in the point-by-point response below

Reviewer 2 Report
Many thanks for your submission. This is a nicely written submission.
The paper reports the authors analysis regards to astragalus in transilvania: the aim is well described, the methods are clear and the results correctly reported.
Conclusion are too extensive and long: I suggest to rehprase the conclusion and report only significative innovation of this paper.
Author Response
We would like to kindly express our appreciation for your time and effort in reviewing the manuscript. We implemented the comments and recommendations as described in the point-by-point response below

Round 2
Reviewer 1 Report
Thank you for responding exhaustively to the requests made